# Generalization and application of the flux-conservative thermodynamic equations in the AROME model of the ALADIN system

Daan Degrauwe[1], Yann Seity[2], François Bouyssel[2], and Piet Termonia[1,3]

[1]RMI Belgium, Ringlaan 3, 1180 Ukkel, Belgium
[2]CNRM, Météo-France, Avenue Coriolis 42, Toulouse, France
[3]Department of Physics and Astronomy, Ghent University, Proeftuinstraat 86, Ghent, Belgium

*Correspondence to:* Daan Degrauwe (daan.degrauwe@meteo.be)

**Abstract.** General yet compact equations are presented to express the thermodynamic impact of physical parameterizations in a NWP or climate model. By expressing the equations in a flux-conservative formulation, the conservation of mass and energy by the physics parameterizations is a built-in feature of the system. Moreover, the centralization of all thermodynamic calculations guarantees a consistent thermodynamical treatment of the different processes. The generality of this physics-dynamics interface is illustrated by applying it in the AROME NWP model. The physics-dynamics interface of this model currently makes some approximations, which typically consist of neglecting some terms in the total energy budget, such as the transport of heat by falling precipitation, or the effect of diffusive moisture transport. Although these terms are usually quite small, omitting them from the energy budget breaks the constraint of energy conservation. The presented set of equations allows to get rid of these approximations, in order to arrive at a consistent and energy-conservative model. A verification in an operational setting shows that the impact on monthly-averaged, domain-wide meteorological scores is quite neutral. However, under specific circumstances, the supposedly small terms may turn out not to be entirely negligible. A detailed study of a case with heavy precipitation shows that the heat transport by precipitation contributes to the formation of a region of relatively cold air near the surface, the so-called cold pool. Given the importance of this cold pool mechanism in the life-cycle of convective events, it is advisable not to neglect phenomena that may enhance it.

## 1 Introduction

The conservation of mass and energy are important characteristics of a numerical atmospheric model. Especially in view of the application in climate studies, even small violations of the conservation laws can accumulate over a long integration time, and lead to faulty results (Staniforth and Wood, 2008; Lucarini and Ragone, 2011). Atmospheric forecast models are usually constructed by combining a dynamical core with physical parameterizations. In general, the dynamical core describes the atmospheric behaviour up until the resolved scales, while the physical parameterizations estimate the effect of subgrid processes (Gassmann, 2013).

A lot of research has been spent in designing dynamical cores that conserve mass and energy (Thuburn, 2008). Common strategies include a careful selection of the prognostic variables (Ooyama, 1990, 2001; Klemp et al., 2007), the formulation of the equations in flux-form (Satoh, 2003), or taking advantage of properties of the Hamiltonian character of the atmospheric equations (Salmon, 2004; Gassmann and Herzog, 2008; Zängl et al., 2014). In contrast with these efforts on the dynamical core, the energy conservation and consistent thermodynamics seem to be less of a priority in the development of the physical parameterizations, or in the way they are coupled to the dynamical core.

A possible explanation is that the thermodynamics of the dynamical core are less complicated than those of the physical parameterizations. More specifically, the dynamics are usually considered adiabatic and reversible (except for numerical diffusion) (Gassmann, 2013). The physics parameterizations, on the other hand, include mass and energy exchange with the surface, as well as radiative fluxes at the top of the atmosphere. They constitute an open thermodynamic system, for which the conservation laws are more difficult to enforce. Moreover, it is tempting to consider physics parameterizations as plug-compatible, i.e. they are considered as a black box which, given an atmospheric state, returns an effect on the dynamical prognostic variables. Unfortunately, this plug-compatibility seems to go at the expense of carefully investigating the thermodynamic consistency between the dynamical core and the physics parameterizations, and inserting a new parameterization in a model comes with implicit assumptions and ad-hoc approximations.

There is, however, an increased interest in different aspects of the coupling of physical parameterizations to the dynamical core. One of the issues is the organization of the timestep. This problem has been studied with academic toy-models (see, e.g. Caya et al. (1998); Staniforth et al. (2002); Termonia and Hamdi (2007)), as well as in 3D models (Hortal, 2002; Williamson, 2002). The thermodynamic aspects of the physics-dynamics coupling is another topic that deserves some attention. Although some attempts have been made to rigorously formulate the equations for a multicomponent atmosphere (Ooyama, 2001; Bannon, 2002), it remains a fact that many operational models make several ad-hoc approximations (Bryan and Fritsch, 2002). Catry et al. (2007), hereafter CGTBT07, presented a set of equations that express the effects of physics parameterizations in a flux-conservative formulation. The advantage of this approach is that this system is inherently mass- and energy-conservative.

The current paper develops the proposal of CGTBT07 further by generalizing it for a system with an arbitrary number of hydrometeors with arbitrary interactions between them. It should be emphasized that the scope of this work is limited to the coupling of the atmospheric physics parameterizations to the dynamical core. For instance, when energy-conserving equations are presented, this property does not necessarily hold for the atmospheric model as a whole, but only regarding the influence of the physical parameterizations. Other aspects of the model, most notably its dynamical core, may not be energy-conserving. Also the mutual interactions between different parameterizations are not considered in this paper, as they relate only indirectly to the time evolution of the prognostic atmospheric variables. The next section presents the equations of this generalized system. In section 3, this set of equations is applied in the AROME numerical weather prediction (NWP) model (Seity et al., 2011), thus allowing to get rid of some approximations that are currently made. Section 4 discusses the impact on the meteorological results, both by means of monthly scores and with an in-depth case study of a cold pool formation under heavy precipitation. Section 5 presents the conclusions.

## 2 Formulation of the generalized flux-conservative equations

### 2.1 Framework of hypotheses

Because the behaviour of the atmosphere is too complex to be described exactly, every numerical model needs to make simplifying hypotheses. This is no different for the work described in the current paper. It is not our aim to present a set of equations which is exact in the sense that it is free of approximations. But a crucial aspect of the work presented in CGTBT07, is that the set of hypotheses that relate to the thermodynamics, is defined from the very beginning. This is important for two reasons. First, it ensures that the simplifications act consistently throughout the model. Second, it allows to set some non-negociable constraints. For instance, the conservation of energy must be satisfied, no matter what other simplifications are made. This approach of setting the simplifying hypotheses from the beginning contrasts with the conventional approach of ignoring supposedly small terms along the way.

The framework of hypotheses is the following:

– A fully barycentric view of air parcels is adopted. This means that all hydrometeors (both suspended and precipitating) are considered as integral parts of the air, and contribute to the parcel's motion, density and heat capacity. This barycentric view has been studied and motivated by many researchers (Wacker and Herbert, 2003; Bott, 2008; Gassmann and Herzog, 2008).

– Water condensates are assumed to have zero volume. This is a common approximation in atmospheric modeling.

– Gases follow Boyle-Mariotte's and Dalton's laws.

– Temperature is homogeneous across all species, even falling hydrometeors. For small hydrometeors, this approximation is easily justified, given their short relaxation time (Bott, 2008). For larger hydrometeors, it is a cruder approximation, but it goes together with the barycentric view: since such hydrometeors are considered part of the parcel, they also take the parcel's temperature.

– The specific heat values of all species are constant with temperature.

– The latent heat values of sublimation and evaporation, $L_i$ and $L_l$ respectively, vary linearly with temperature $T$:

$$L_{i|l}(T) = L_{i|l}(T_0) + (c_{pv} - c_{i|l})T \tag{1}$$

with $T_0 = 0$K, $c_{pv}$ is the specific heat capacity at constant pressure of water vapour, and $c_i$ and $c_l$ are the specific heat capacity values of ice and liquid water, respectively.

It should be mentioned that this same framework of assumptions has been used by Marquet (2011), Marquet and Geleyn (2013) and Marquet (2015) to cleanly develop moist atmospheric thermodynamic quantities such as moist entropy, moist potential temperature, and moist Brunt-Väisälä frequency.

## 2.2 The flux-conservative equations for a system with 5 water species

The system considered in CGTBT07 consists of dry air (specific mass fraction $q_d = \rho_d/\rho_{tot}$) plus five prognostic water species: vapour (specific mass fraction $q_v$), suspended liquid water droplets ($q_l$), suspended ice crystals ($q_i$), precipitating rain ($q_r$) and precipitating snow ($q_s$). For this system, the following equations are derived for the time evolution of the prognostic species due to physical parameterizations:

$$\frac{\partial q_v}{\partial t} = g\frac{\partial}{\partial p}\left[ R_{r,v} + R_{s,v} - R_{v,l} - R_{v,i} + \frac{q_v(P_r + P_s)}{1 - q_r - q_s} - D_v \right] \tag{2}$$

$$\frac{\partial q_l}{\partial t} = g\frac{\partial}{\partial p}\left[ R_{v,l} - R_{l,r} + \frac{q_l(P_r + P_s)}{1 - q_r - q_s} - D_l \right] \tag{3}$$

$$\frac{\partial q_r}{\partial t} = g\frac{\partial}{\partial p}\left[ R_{l,r} - R_{r,v} - P_r \right] \tag{4}$$

$$\frac{\partial q_i}{\partial t} = g\frac{\partial}{\partial p}\left[ R_{v,i} - R_{i,s} + \frac{q_i(P_r + P_s)}{1 - q_r - q_s} - D_i \right] \tag{5}$$

$$\frac{\partial q_s}{\partial t} = g\frac{\partial}{\partial p}\left[ R_{i,s} - R_{s,v} - P_s \right] \tag{6}$$

$$\frac{\partial q_d}{\partial t} = g\frac{\partial}{\partial p}\left[ \frac{q_d(P_r + P_s)}{1 - q_r - q_s} - D_d \right] \tag{7}$$

In these equations, $P_k$ denotes precipitation fluxes, and $D_k$ denotes diffusive fluxes. Note that it is necessary that $\sum_{k=d,v,i,l} D_k = 0$ to ensure that all terms on the right hand sides cancel out. The terms $R_{k_1,k_2}$ denote pseudofluxes and represent mass transfer between two water species. The concept of pseudofluxes is essential to the presented system and deserves some more explanation. The common and more intuitive way to express a mass transfer between two species is through a time tendency. For instance, consider the microphysical process of condensation, which is a mass transfer from water vapour to liquid cloud water droplets. The effect of this process on the specific humidities could be expressed as

$$\frac{\partial q_v}{\partial t} = -\left(\frac{\partial q}{\partial t}\right)^{cond}$$

$$\frac{\partial q_l}{\partial t} = \left(\frac{\partial q}{\partial t}\right)^{cond}$$

The pseudoflux $R_{v,l}$ expresses exactly the same effect, only as a flux instead of as a tendency. This flux is determined by taking the vertical integral of the tendency:

$$R_{v,l} = \frac{1}{g}\int_0^p \left(\frac{\partial q}{\partial t}\right)^{cond} dp$$

Although a pseudoflux is arguably more difficult to interpret than a tendency, writing conversions between species in terms of pseudofluxes offers the possibility to write the evolution equations in a flux-conservative form. The benefit of this is explained further. Also note that this does not mean that the internals of the physics parameterizations should be formulated in terms of pseudofluxes. Instead, it is only at the moment when the contributions of the physics parameterizations are added to the prog-

nostic variables, that pseudofluxes are beneficial. They can be determined at that point from the more conventional tendencies using the expression above.

The thermodynamic equation for the system with 4 hydrometeors is as follows:

$$\frac{\partial}{\partial t}(c_p T) = -g\frac{\partial}{\partial p}\big[(c_l - c_{pd})P_r T + (c_i - c_{pd})P_s T$$

$$- (\hat{c} - c_{pd})(P_r + P_s)T + J_s + J_{rad}$$

$$- L_l(T_0)(R_{v,l} - R_{r,v}) - L_i(T_0)(R_{v,i} - R_{s,v})\big] \tag{8}$$

where $\hat{c} = \frac{c_{pd}q_d + c_{pv}q_v + c_i q_i + c_l q_l}{1 - q_r - q_s}$, and $J_s$ and $J_{rad}$ are the diffusive and radiative heat fluxes, respectively. $c_p$ is the total heat capacity of the parcel, given by

$$c_p = c_{pd}q_d + c_{pv}q_v + c_l(q_l + q_r) + c_i(q_i + q_s)$$

It should be noted that Eq. (8) expresses only the thermodynamic effect of the physical parameterizations. The complete thermodynamic equation of the atmospheric model would also include terms that are resolved by the dynamics of the model. A full discussion of these equations is given in CGTBT07, but we would like to stress the following characteristics:

- All equations are flux-conservative, i.e. every right hand side is a divergence of a summation of fluxes. The importance of this property cannot be underestimated, because it means that this system intrinsically conserves mass and energy. Put somewhat simplistically, in a flux-conservative system, the only way energy or mass can leave one model layer, is by transporting it to an adjacent layer. Therefore, mass and energy are conserved by design of the system.

- The precipitation fluxes $P_r$ and $P_s$ are relative to the (moving) center of mass of the parcel. They relate to the absolute precipitation fluxes $P_r^*$ and $P_s^*$ through

$$P_r = (1 - q_r)P_r^* - q_r P_s^* \tag{9}$$

$$P_s = -q_s P_r^* + (1 - q_s)P_s^* \tag{10}$$

To derive these relations, one starts from the definition of a flux as a product of a density with a velocity. For instance, for rain, one writes

$$P_r^* = \rho_{tot}q_r w_r^*$$

The absolute velocity $w^*$ of the center of mass of the parcel is given by the weighted average of the velocities of the components. In a system where only rain and snow are precipitating, this means that

$$w^* = q_r w_r^* + q_s w_s^*$$

The relative velocity of rain is then given by $w_r = w_r^* - w^*$, so the relative precipitation flux becomes

$$P_r = \rho_{tot}q_r w_r = \rho_{tot}(q_r w_r^* - q_r^2 w_r^* - q_r q_s w_s^*) = (1 - q_r)P_r^* - q_r P_s^*$$

- The latent heat values of sublimation and condensation $L_i$ and $L_l$ that appear on the right hand side, are evaluated at $T_0 = 0\mathrm{K}$. This does not mean that the temperature-dependency of these latent heat values is neglected. Instead, it is accounted for by considering the time derivative of the enthalpy $c_p T$. Considering only the process of condensation, the traditional way to express its thermodynamic effect would be

$$c_p \frac{\partial T}{\partial t} = L_l(T) \frac{\partial q_l}{\partial t}$$

Using the before-set assumption that $L_l$ varies linearly with temperature, and the fact that, still only considering condensation, $\partial q_v / \partial t = -\partial q_l / \partial t$, so $\partial c_p / \partial t = (c_l - c_{pv}) \partial q_l / \partial t$, this expression becomes

$$c_p \frac{\partial T}{\partial t} = L_l(T_0) \frac{\partial q_l}{\partial t} - T \frac{\partial c_p}{\partial t}$$

which can be rewritten as

$$\frac{\partial}{\partial t}(c_p T) = L_l(T_0) \frac{\partial q_v}{\partial t}$$

This shows how the temperature-dependence of the latent heat values can be accounted for by considering the tendency of enthalpy.

- Although the equations only describe the evolution of water species, similar flux-conservative equations could be formulated for other atmospheric variables like momentum, turbulent kinetic energy, etc. In this manuscript, only water species and their effect on the thermodynamic equation are studied.

## 2.3 The generalized flux-conservative equations

Despite the clear strength of the equations proposed by CGTBT07, their application is not straightforward because of the fixed number of water species, and because of the fixed set of interactions between them (six pseudofluxes). More advanced microphysics schemes often consider more water species, for instance by including graupel and/or hail (Lascaux et al., 2006), or by separating convective and nonconvective fractions of hydrometeors (Piriou et al., 2007). Also the fact that only six transfer mechanisms between the water species are possible is limiting. For instance, snow melting cannot be represented directly, but it should be written as a combination of snow sublimation ($R_{s,v}$) and rain evaporation ($R_{r,v}$). Although thermodynamically fully correct, it would be better to have a system that digests all kinds of transfers between water species.

It is, however, possible to generalize the equations from CGTBT07, without touching the important characteristics. We introduce the following notation: $n$ is the number of water species, the index $k = 1, \ldots, n$ denotes a single water species, and by convention, $k = 0$ denotes the dry air component. The specific heat capacity values at constant pressure of the different species are written generically as $c_k$, the latent heat of evaporation or sublimation at $0\,K$ is written as $L_k^0$. The index $j$ denotes a conversion process between a source water species $k_j^s$ and a target water species $k_j^t$. The effect of this process is expressed through the pseudoflux $R_j$. We consider an arbitrary number $m$ of such conversion processes. We now define variables $\lambda_{kj} = \delta_{k,k_j^s} - \delta_{k,k_j^t}$ for $k = 1, \ldots, n$ and for $j = 1, \ldots, m$, where the usual definition of the Kronecker delta is used. This variable takes

**Table 1.** Variables $\lambda_{kj}$ for the system of CGTBT07

| | | process | $r \to v$ | $v \to l$ | $l \to r$ | $s \to v$ | $v \to i$ | $i \to s$ |
|---|---|---|---|---|---|---|---|---|
| | | $j$ | 1 | 2 | 3 | 4 | 5 | 6 |
| species | $k$ | | | | | | | |
| $v$ | 1 | | -1 | 1 | 0 | -1 | 1 | 0 |
| $l$ | 2 | | 0 | -1 | 1 | 0 | 0 | 0 |
| $r$ | 3 | | 1 | 0 | -1 | 0 | 0 | 0 |
| $i$ | 4 | | 0 | 0 | 0 | 0 | -1 | 1 |
| $s$ | 5 | | 0 | 0 | 0 | 1 | 0 | -1 |

**Table 2.** Variables $\Lambda_j^0$ for the system of CGTBT07

| process | $r \to v$ | $v \to l$ | $l \to r$ | $s \to v$ | $v \to i$ | $i \to s$ |
|---|---|---|---|---|---|---|
| $j$ | 1 | 2 | 3 | 4 | 5 | 6 |
| $\Lambda_j^0$ | $L_l(T_0)$ | $-L_l(T_0)$ | 0 | $L_i(T_0)$ | $-L_i(T_0)$ | 0 |

value 0 if a species $k$ is not involved in the conversion process $j$; it takes value -1 if it is the target species of this process, and it takes value 1 if it is the source species of this process. The variable $\lambda_{kj}$ will allow to write the time tendency of a water species by summing over all conversion processes, regardless of the role this specific water species plays in each process. Furthermore, a variable $\Lambda_j^0 = L_{k_j^s}^0 - L_{k_j^t}^0$ is defined. This variable is the latent heat released at temperature $T_0$ under a conversion process

with source species $k_j^s$ and target process $k_j^t$. To clarify these notations, consider the original system of CGTBT07 with 5 water species and 6 conversion processes between them. By convention, we assign $k = 1, \ldots, 5$ to water vapour, liquid cloud water, precipitating rain, cloud ice crystals and precipitating snow, respectively. Tables 1 and 2 give the values of $\lambda_{kj}$ and $\Lambda_j^0$ for the different conversion processes.

Next, a precipitation flux $P_k$ is defined for each component, even for the non-precipitating species (dry air, vapour, liquid

cloud water droplets and cloud ice crystals). Contradictory as this may sound, it should be stressed that in our barycentric system, these fluxes express the motion of the species with respect to the center of mass of the parcel. When precipitating species are present, the suspended species will move upward with respect to the mass center. Using a similar calculation as before to describe the motion with respect to the center of mass of the parcel, the relative precipitation fluxes $P_k$ are determined from the absolute fluxes $P_k^*$ as

$$P_k = P_k^* - q_k \sum_{i=0}^{n} P_i^* \tag{11}$$

where the absolute precipitation fluxes of suspended species can be taken to be zero. It should be noted that the strict distinction in CGTBT07 between suspended and precipitating species is somewhat arbitrary and scale-dependent. Indeed, also the so-

called suspended cloud water species can undergo a slow sedimentation. This arbitrary distinction is no longer necessary in the generalized set of equations that is presented here. Similarly to defining (relative) precipitation fluxes for all species, also diffusive fluxes $D_k$ are defined for all species, where the diffusive fluxes of precipitating species can be taken equal to zero.

These notations make it possible to formulate the specific mass equations and the thermodynamic equation as follows:

$$\frac{\partial q_k}{\partial t} = -g \frac{\partial}{\partial p} \left[ \sum_{j=1}^{m} \lambda_{kj} R_j + P_k + D_k \right], \qquad \text{for } k = 0, \ldots, n \tag{12}$$

$$\frac{\partial}{\partial t}(c_p T) = -g \frac{\partial}{\partial p} \left[ \sum_{k=0}^{n} c_k P_k T + J_s + J_{rad} - \sum_{j=1}^{m} \Lambda_j^0 R_j \right] \tag{13}$$

These equations generalize the ones from CGTBT07 in three ways: (i) an arbitrary number $n$ of water species is considered; (ii) an arbitrary number $m$ of inter-species conversion processes is considered; and (iii) the strict distinction between suspended and precipitating species can be abandoned. The fact that quite compact equations are obtained, which are valid for all components of the atmosphere, is an additional indication of the strength of the barycentric approach.

## 2.4 Remarks

Some comments should be given on the application area of the physics-dynamics interface presented in Eqs. (12–13).

– The fact that these equations are very general, opens the road for a 'plug-compatible' view of physics parameterizations. Indeed, the only output that is needed from a parameterization are diffusive and precipitative transport fluxes, pseudofluxes for phase changes and the radiative and diffusive energy fluxes. The physics-dynamics interface then receives these quantities and determines the effect on the prognostic variables of the model, thereby ensuring satisfaction of the conservation of mass and energy, as well as consistency in the thermodynamic assumptions.

However, it should be kept in mind that other conditions should be met before parameterizations can really be considered plug-compatible. A first aspect is that interactions exist between parameterizations. For instance, the parameterization of cloud processes will affect the radiation scheme. This kind of interactions should properly be accounted for when plugging a new parameterization into a model. In this context, it is interesting to see that the technical recommendations that were made in Kalnay et al. (1989) regarding the design of parameterizations and their interactions, are still relevant at present. A second aspect is that parameterizations should also obey the second law of thermodynamics (Gassmann and Herzog, 2014). This condition cannot be enforced at the higher level of the physics-dynamics interface, and should be taken care of at the level of the parameterization itself.

– A common assumption in atmospheric modeling (although it is often made implicitly) is that all vertical mass transport due to the physics parameterizations is compensated for by a fictitious flux of dry air (Courtier et al., 1991). This assumption ensures the conservation of total mass in the atmosphere, but makes it impossible to express a net mass exchange with the surface due to for instance precipitation. From a barycentric point of view, this approximation means that the

center of mass of an air parcel does not move vertically. The equations (12–13) remain valid under this assumption, if the relative flux of dry air is defined as $P_d = -\sum_{k=1}^{n} P_k$.

– The Eqs. (12–13) are theoretically only valid for a model using the hydrostatic primitive equations. In a fully compressible system, the diabatic heating from the physics parameterizations does not only affect the temperature equation, but also the continuity equation (Laprise, 1998). CGTBT07 present the extension of their flux-conservative system to the fully compressible case. An entirely equivalent development can be made for the generalized equations presented in this paper.

However, as shown by Malardel (2010), the impact of including the heat from parameterizations as a forcing in the continuity equation is quite limited. In other words, one can apply the thermodynamic equation (13) also in a non-hydrostatic model.

– The fact that Eq. (13) describes the evolution of enthalpy $h = c_p T$, does not mean that this variable should become the prognostic thermodynamic variable of the model. A model that uses temperature $T$ as the prognostic thermodynamic variable, can also use the presented interface. After all, one can easily calculate the total heat capacity tendency as follows:

$$\frac{\partial c_p}{\partial t} = \sum_{k=0}^{n} c_k \frac{\partial q_k}{\partial t}, \tag{14}$$

which in turn can be used to determine the temperature tendency from the enthalpy tendency:

$$\frac{\partial T}{\partial t} = \frac{1}{c_p} \left( \frac{\partial}{\partial t}(c_p T) - T \frac{\partial c_p}{\partial t} \right) \tag{15}$$

The importance of writing Eq. (13) as a time evolution of enthalpy only becomes clear in the time-discretized case.

$$c_p^{t+\Delta t} = c_p^t + \Delta t \sum_{k=0}^{n} c_k \frac{\partial q_k}{\partial t} \tag{16}$$

$$T^{t+\Delta t} = \frac{1}{c_p^{t+\Delta t}} \left( c_p^t T^t + \Delta t \frac{\partial}{\partial t}(c_p T) \right) \tag{17}$$

where a superscript $t$ denotes variables at the current timestep, while a superscript $t + \Delta t$ denotes variables at the next timestep. Using an enthalpy-based formulation of the interface is reflected in the use of $c_p^{t+\Delta t}$ in the right-hand side of Eq. 17. Although this appears to be a small detail, it is crucial in ensuring the conservation of energy. The importance of appropriately discretizing a conserved nonlinear variable such as enthalpy is also indicated by Gassmann and Herzog (2008).

As a side remark, it can be noted that simply adding temperature tendencies from several parameterizations cannot lead to an energy-conserving atmospheric model, at least not for a process-split coupling strategy (Williamson, 2002). For example, consider a model containing two parameterizations (indicated with $a$ and $b$), yielding a respective change in temperature of $\Delta T^a$ and $\Delta T^b$, and a respective change in heat capacity of $\Delta c_p^a$ and $\Delta c_p^b$. Suppose that each of

these parameterizations is energy-conservative in itself, meaning that the enthalpy changes are respectively $\Delta h^a = (c_p^t + \Delta c_p^a)(T^t + \Delta T^a) - c_p^t T^t$ and $\Delta h^b = (c_p^t + \Delta c_p^b)(T^t + \Delta T^b) - c_p^t T^t$. Then the joint effect of the parameterizations cannot be expressed as $\Delta T = \Delta T^a + \Delta T^b$, but it should be determined as

$$\Delta h = \Delta h^a + \Delta h^b, \tag{18}$$

from which the total change in temperature is determined as

$$\Delta T = \frac{(c_p^t + \Delta c_p^a)\Delta T^a + (c_p^t + \Delta c_p^b)\Delta T^b}{c_p^t + \Delta c_p^a + \Delta c_p^b}$$

This expression is only valid for a process-split coupling. For a time-split coupling, the total enthalpy change is still equal to the sum of the enthalpy changes of the separate processes, as indicated in Eq. (18). However, the fact should should be taken into account that process $b$ does not start from $c_p^t$ and $T^t$, but rather from the atmospheric state after accounting for process $a$, i.e. $\tilde{c}_p = c_p^t + \Delta c_p^a$, and $\tilde{T} = T^t + \Delta T^a$. So for a time-spit coupling, the enthalpy change of process $b$ becomes

$$\Delta h^b = (\tilde{c}_p + \Delta c_p^b)(\tilde{T} + \Delta T^b) - \tilde{c}_p \tilde{T}$$

Working out the heat capacity and the temperature at the end of the timestep now gives

$$c_p^{t+\Delta t} = c_p^t + \Delta c_p^a + \Delta c_p^b$$
$$T^{t+\Delta t} = \frac{c_p^t T^t + \Delta h}{c_p^{t+\Delta t}} = T^t + \Delta T^a + \Delta T^b$$

So with time-split coupling, the total temperature change can be obtained as the summation of the temperature changes from the separate parameterizations. However, it is better to use an enthalpy-based system, as this works both for the process-split and the time-split cases.

– The Eqs. (12)–(13) only describe the evolution of the atmospheric prognostic variables. The prognostic variables of the surface scheme are not part of this system. In this context, the work of Best et al. (2004) should be mentioned. They present a method to separate the surface scheme from the atmospheric model. The core of this method is to describe the interaction between atmosphere and surface with fluxes. In this sense, their work matches perfectly with the flux-based equations (12)–(13).

## 3  Application of the flux-conservative equations in the AROME model

AROME is a limited area model that was developed at Météo-France and is now a configuration inside the ALADIN system. It became operational in France in 2008, and it is currently used in many European countries of the ALADIN and HIRLAM consortia. AROME uses a nonhydrostatic, fully compressible dynamical core (Bubnová et al., 1995; Bénard et al., 2010), with the same spectral semi-implicit semi-Lagrangian space-time discretization as the ECMWF's IFS model. The height coordinate

is terrain-following mass-based (Laprise, 1992). The physics parameterizations in AROME originate from the Méso-NH research model (Lafore et al., 1998). The Méso-NH model has a dynamical core which is explicit in time, with a staggered spatial grid and a height-based vertical coordinate, so it is substantially different from the AROME dynamical core. The plugging of the physics from this model to a different dynamical core was quite challenging, and several approximations were made during this process.

A first approximation that is made in the existing AROME physics-dynamics interface concerns the heat transport by precipitation. From Eq. (13), it is clear that precipitation has two thermodynamic effects. Falling species modify the composition of the atmosphere, so they also change the specific heat capacity $c_p = \sum_{k=0}^{n} c_k q_k$. Secondly, if a vertical temperature gradient exists, falling species are heated, thus cooling down the surrounding air. The effect on the enthalpy due to a change in $c_p$ is given by

$$\left(\frac{\partial}{\partial t} c_p T\right)^{prec,c_p} = T \sum_{k=0}^{n} c_k \frac{\partial q_k}{\partial t} = -gT \frac{\partial}{\partial p} \sum_{k=0}^{n} c_k P_k \tag{19}$$

while the second effect due to a vertical temperature gradient is given by

$$\left(\frac{\partial}{\partial t} c_p T\right)^{prec,heat} = -g \sum_{k=0}^{n} c_k P_k \frac{\partial T}{\partial p} \tag{20}$$

The combination of these two effects indeed corresponds to the effect of precipitation in the right-hand side of Eq. (13):

$$\left(\frac{\partial}{\partial t} c_p T\right)^{prec,c_p} + \left(\frac{\partial}{\partial t} c_p T\right)^{prec,heat} = -g \frac{\partial}{\partial p} \sum_{k=0}^{n} c_k P_k T \tag{21}$$

The approximation made by the existing physics-dynamics interface in AROME is that it neglects the heat transport effect of precipitation, i.e. the term given in Eq. (20).

A second approximation concerns the effect of diffusive moisture transport (shallow convection and turbulence) in the energy budget. Similar to the effect of precipitation, diffusive moisture transport modifies the total specific heat capacity $c_p$, and this effect should be accounted for in the energy budget. However, this effect is neglected in the existing AROME physics-dynamics interface.

A third approximation is that the values of specific heat capacity $c_p$ and latent heat $L_{i|l}(T)$ are not consistent between the different parameterizations. For instance, the heat capacity in the radiation scheme only accounts for water vapour and neglects the other hydrometeors ($c_p^{rad} = (1 - q_v)c_{pd} + q_v c_{pv}$). This situation stems from the fact that the different physics parameterizations are developed by different teams, each using their own conventions.

A final approximation by the existing physics-dynamics interface in AROME is that the total temperature tendency is obtained by summing the temperature tendencies from the individual parameterizations. As indicated in the previous section, such an approach cannot lead to an energy-conserving system.

Although it can be expected that the overall effect of these approximations and inconsistencies is quite limited, the generalized physics-dynamics interface as presented in the previous section offers the possibility to get rid of them in order to take a (admittedly small) step towards a more accurate model. A second motivation to equip the AROME model with the generalized

flux-conservative physics dynamics interface is that this opens the route towards importing physics parameterizations from other NWP models, thus allowing a fair comparison of different parameterizations and stimulating scientific progress.

## 4 Impact on weather forecast

The impact of the presented flux-conservative formulation of the physics-dynamics interface is investigated with the AROME operational high-resolution LAM model running at Météo-France. Before April 2015, this model ran on a $739 \times 709$ grid with a resolution of 2.5 km. Figure 1 shows the model domain. The timestep is 60 s. The model is provided with lateral boundary conditions by the operational global model 'ARPEGE' from Météo-France. The initial conditions are generated with a 3D-Var data-assimilation (Fischer et al., 2005; Brousseau et al., 2011).

At the surface level, precipitation and evapotranspiration imply a net mass-flux across the surface. Since the vertical coordinate of the AROME model is mass-based, correctly accounting for such net mass exchange between atmosphere and surface has far-reaching implications, especially in the surface boundary condition of the nonhydrostatic dynamical core. Currently, this has not been implemented in the dynamical core of the AROME model. Instead, the above-mentioned approximation is made that all vertical transport due to the parameterizations is compensated by a fictitious flux of dry air. Taking full advantage of the barycentric framework of Eqs. (12–13) would require an adaptation of the dynamical core of AROME, which falls outside the scope of this work.

All these settings are identical for the operational run (denoted REF) with the temperature-tendency based interface and for the run with the flux-conservative interface (denoted FCI).

### 4.1 Monthly scores

The daily forecasts during two periods are considered in this section: 1–30 November 2014 and 6 January – 6 February 2015. The first month is characterized by exceptionally mild weather, with numerous episodes of heavy precipitation in the South-West of France. The second month was characterized by strong winds and episodes of heavy snowfall. Figures 2 and 3 show bias and rmse for several meteorological variables for the two periods, respectively. These scores are calculated by comparing the AROME forecasts with observations throughout the French territory. Figures 4 and 5 compare the forecasted precipitation over the two periods. To avoid the problem of the double penalty, the precipitation is verified with the neighbourhood observation Brier skill score (Amodei and Stein, 2009). This score is determined by calculating the probability that a precipitation threshold is exceeded in the vicinity of an observation. By choosing the threshold, one focuses the verification more on light or on heavy precipitation.

The scores indicate that the impact of using the flux-conservative set of equations appears quite limited when considering time- and space-averaged scores as the ones considered here. It should be stressed that no retuning has been done for the experiments with the flux-conservative equations. As a result, compensating errors can be responsible for masking an improvement of the scores. The fact that the scores do not change substantially, merely indicates that the approximations that are made in the existing temperature-tendency based interface are indeed small on a domain-wide scale. In this context, the limitations of

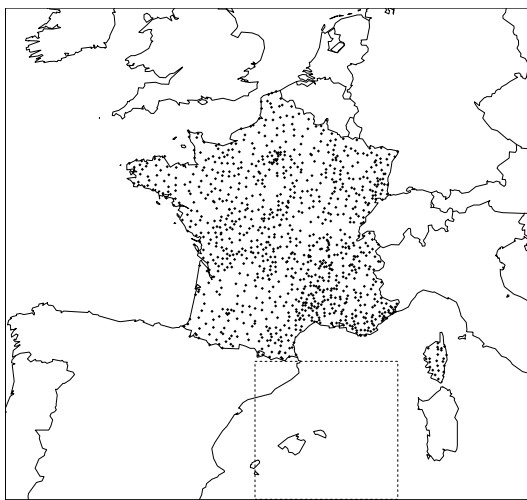

**Figure 1.** Operational AROME domain with a resolution of 2.5 km. The markers indicate the temperature stations used for the monthly scores. The dashed line indicates the area of the case-study of section 4.2.

this standard verification against station data should also be mentioned. By taking the average score over a large number of stations, important local differences may be hidden in the scores. In a similar way, the fact that monthly averaged scores are considered, only allows to detect differences that are systematic in time. Therefore, notwithstanding the neutral impact on the standard scores, some significant differences are observed under specific circumstances. A case study is presented in the next

section to illustrate this.

### 4.2   Case study of a cold pool originating from heavy precipitation

When precipitation evaporates while falling through unsaturated air, it cools its environment. As such, a region of relatively cool air, the so-called cold pool, originates when heavy, localized precipitation occurs, for instance in precipitating convective systems (Fujita, 1959). It has been shown that the cold pool is in fact a key element in the lifecycle of a such systems. On the

one hand, new convective cells originate at the border of the cold pool and its warmer surroundings, but on the other hand, if the cold pool becomes too strong, it may cut off the supply of warm air to the updraft (Engerer et al., 2008). The cold pool is also accompanied by a meso-scale high pressure area (Fujita, 1959) which plays a crucial role in the wind gusts that go with heavy precipitation. For these reasons, it is no surprise that the representation of the cold pool is also crucial in a NWP model (Engerer et al., 2008; De Meutter et al., 2014).

Although evaporative cooling is the main cause for a cold pool, a second mechanism may enhance it. As precipitation falls from colder layers aloft to hotter layers below, it will be heated by the surrounding air, which in response will cool down

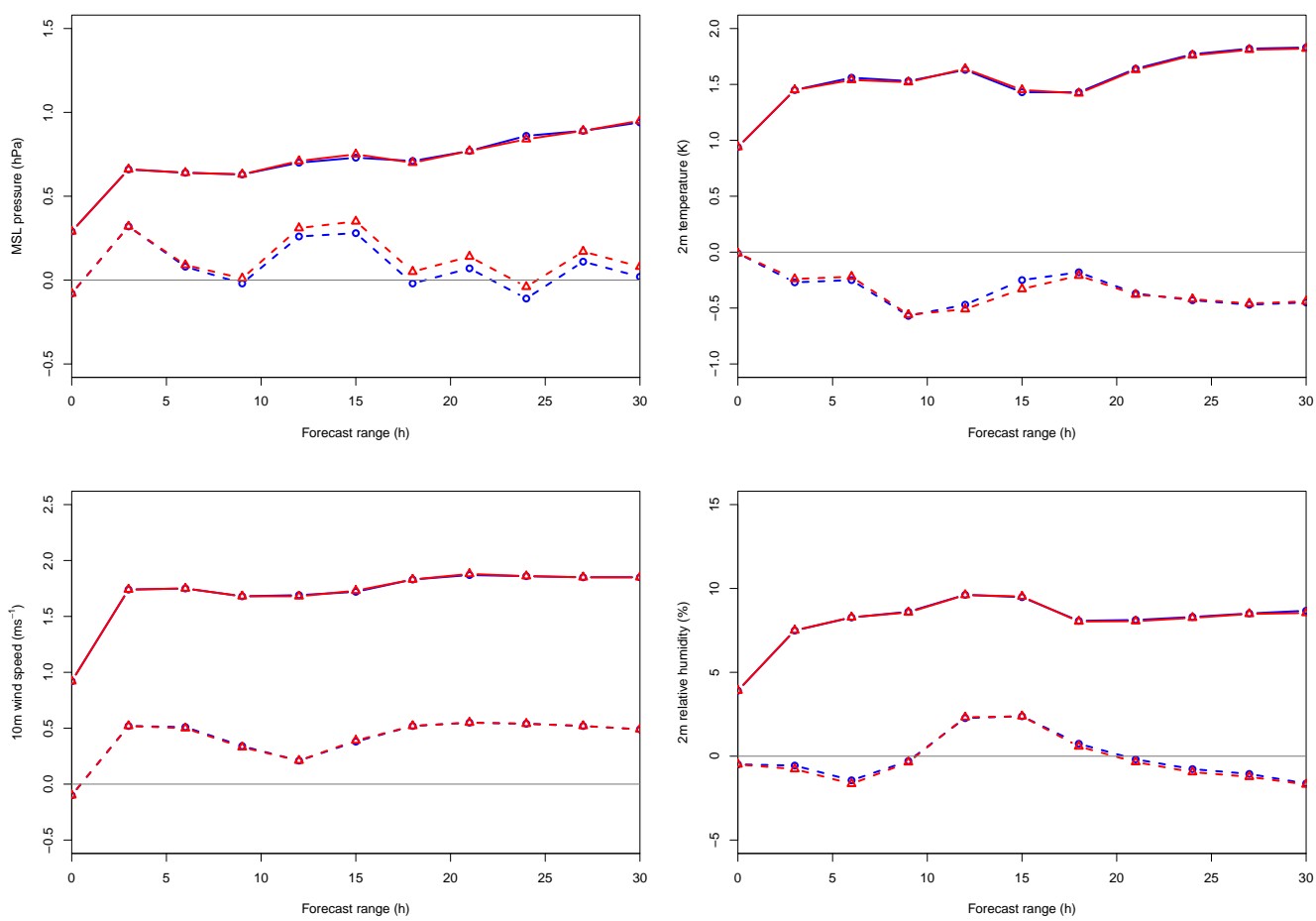

**Figure 2.** RMSE (solid line) and bias (dashed line) over the period 1 November 2014 – 30 November 2014, for REF (blue circles) and FCI (red triangles).

(Johnson and Hamilton, 1988). As explained in Sect. 3, this secondary thermodynamic effect (the transport of sensible heat) of precipitation is neglected in the existing AROME physics-dynamics interface, while it is correctly accounted for with the presented set of flux-conservative equations. One can thus expect that the intensity of a forecasted cold pool depends on which set of equations is used.

5      This is confirmed when looking at the AROME forecasts over the Balearic islands on 19 January 2015. This case is charac­terized by convection developing ahead of an active cold front coming from the south. Figures 6a and 6b show the forecasted 1200UTC–1800UTC accumulated precipitation with the existing AROME interface (REF) and with the flux-conservative inter­face (FCI). It is observed that the overall structure of the precipitation is quite similar. However, when comparing the coldpool characteristics of both experiments, important differences appear. Figures 6c and 6d show the differences between both experi-

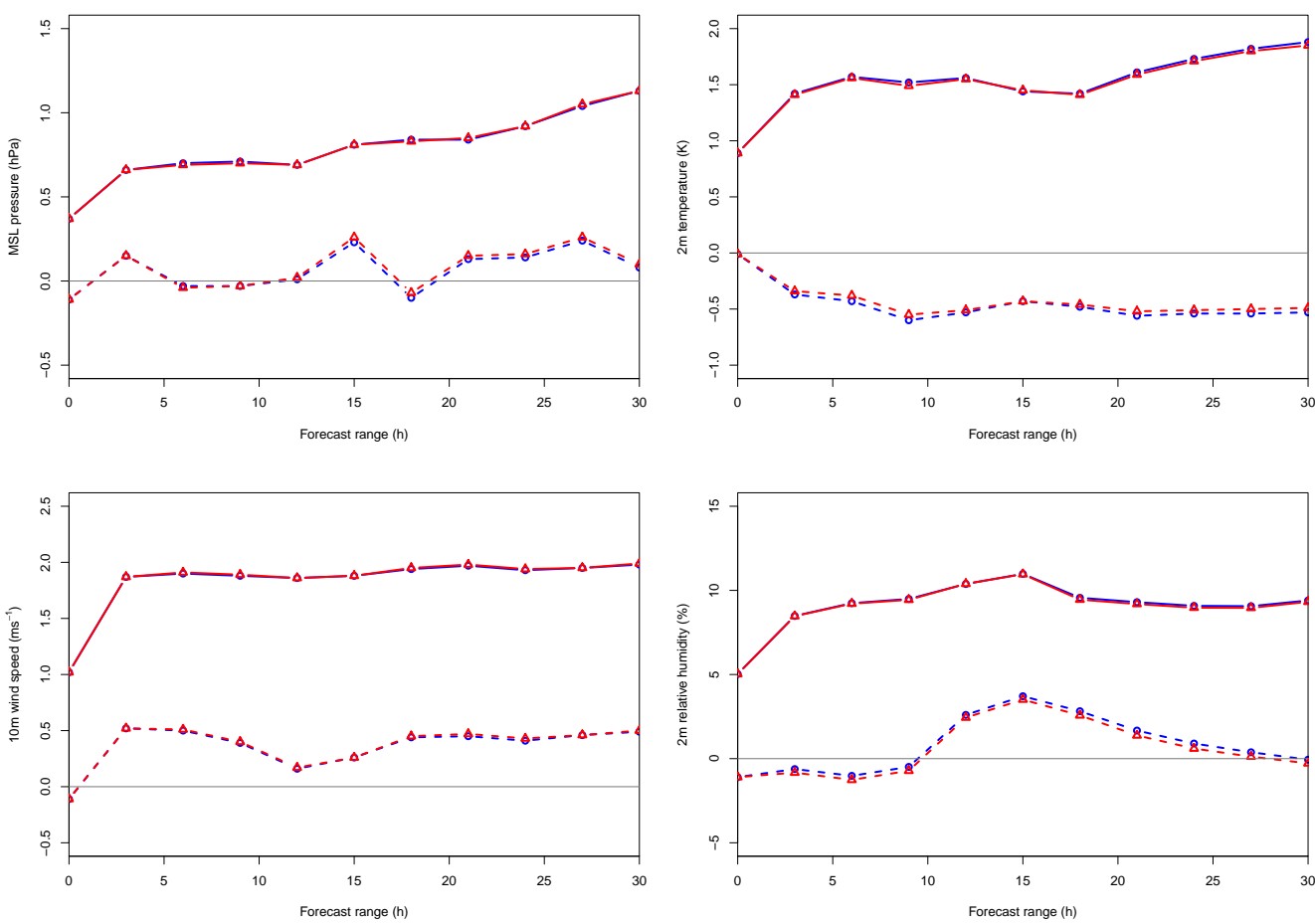

**Figure 3.** RMSE (solid line) and bias (dashed line) over the period 6 January 2015 – 6 February 2015, for REF (blue circles) and FCI (red triangles).

ments for the 2m temperature and the surface pressure. The temperature is significantly lower with FCI (up to 5 K cooler), and the surface pressure is higher (up to 1.4 hPa).

To further illustrate the impact of the heat transport by precipitation on the cold pool, the vertical profiles in the point as marked in figure 6b are studied for the experiment with the flux-conservative interface. The vertical profile of the precipitation fluxes (figure 7a) shows how snow and graupel originate aloft, they melt to form rain at around 850 hPa, and the rain starts to evaporate below 930 hPa. Figure 7b shows the vertical profile of the two phenomena that are responsible for the development of the cold pool, averaged between 1200UTC and 1800UTC: the latent heat effects from phase changes (solid line), and the falling of cold hydrometeors into warmer air layers (dashed line). It is clear that the second effect is orders of magnitude smaller than the first effect, at least when considering the full vertical extent of the model. However, as shown in figure 7c, the heat

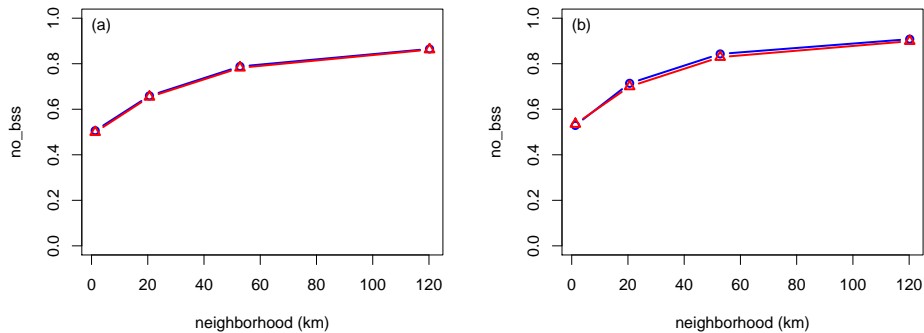

**Figure 4.** Neighbourhood observation Brier skill score for precipitation between 1200UTC and 1800UTC over the period 1 November 2014 – 30 November 2014, for REF (blue circles) and FCI (red triangles): (a) threshold 2 mm; (b) threshold 10 mm

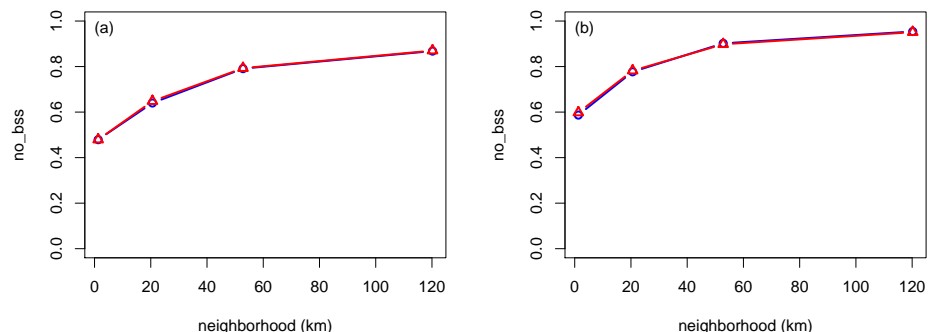

**Figure 5.** Neighbourhood observation Brier skill score for precipitation between 1200UTC and 1800UTC over the period 6 January 2015 – 6 February 2015, for REF (blue circles) and FCI (red triangles): (a) threshold 2 mm; (b) threshold 10 mm

transport by hydrometeors is not entirely negligible in the range between the surface and 900 hPa, and thus contributes to the intensity of the cold pool.

No comparison with observations is done for this case, because the purpose of this case study is merely to illustrate that even small terms can have a significant impact under certain conditions. The conclusions from this case-study are in line with the results from Bryan and Fritsch (2002), where neglecting a supposedly small term in the energy budget unexpectedly leads to the worst results.

## 5   Conclusions

This paper starts from the equations presented in Catry et al. (2007) that describe how the effect of physical parameterizations on the dynamical core of an NWP model can be expressed in a flux-conservative way. The main advantage of these equations

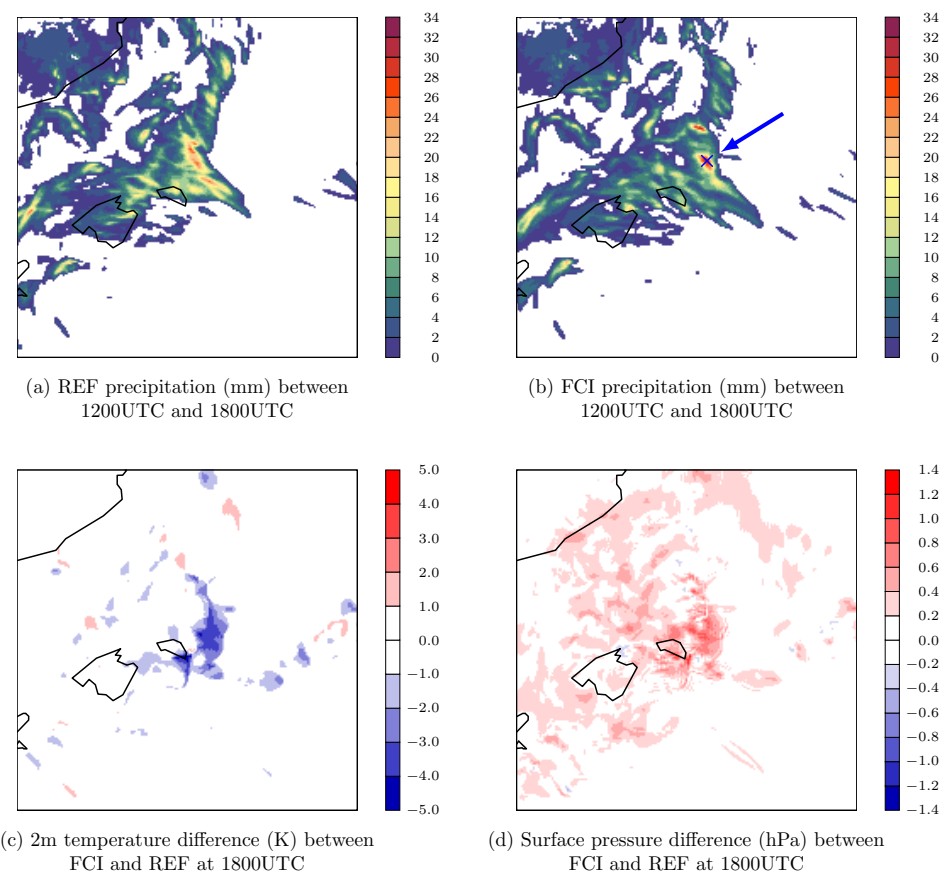

(a) REF precipitation (mm) between
1200UTC and 1800UTC

(b) FCI precipitation (mm) between
1200UTC and 1800UTC

(c) 2m temperature difference (K) between
FCI and REF at 1800UTC

(d) Surface pressure difference (hPa) between
FCI and REF at 1800UTC

**Figure 6.** Case of heavy precipitation on 19 January 2015. The arrow and the marker in subfigure (b) indicate the location of the profiles of Figure 7.

is that they impose the constraints of energy- and mass-conservation at a higher level in the model than at the level of the individual physical parameterizations. The presented equations only guarantee conservation of mass and energy regarding the effect of the physics contributions, not for the dynamical core of the model. A second advantage of the presented equations is that by gathering the thermodynamic calculations of all physics parameterizations in a single equation, it is also guaranteed that a predefined framework of hypotheses is consistently respected.

Notwithstanding these clear advantages, the equation set in the mentioned paper also faces limitations that hinder its application in existing NWP models. This paper presents a generalized set of thermodynamic equations that overcomes these restrictions without touching the sound theoretical foundations. More specifically, the presented equations are valid for an arbitrary number of hydrometeors, and can be applied in a model with an arbitrary number of conversion processes between these water species. This has allowed to use this set of equations in the AROME NWP model, which currently uses a physics-

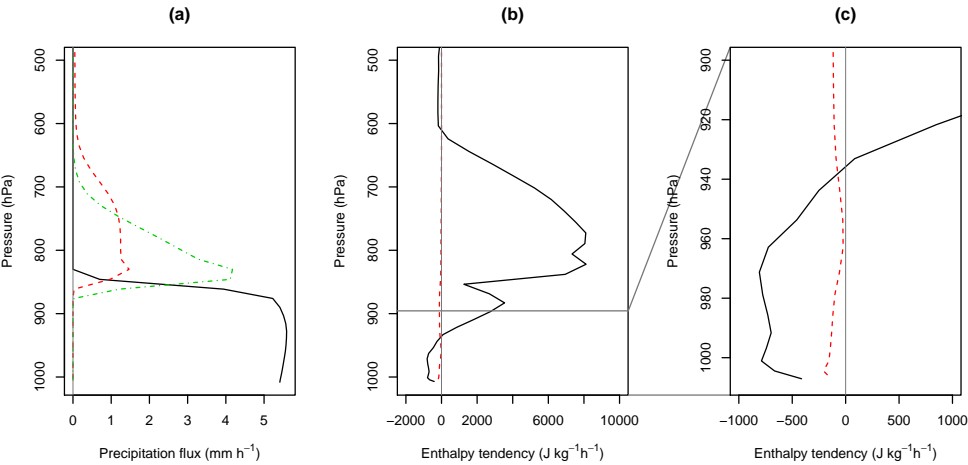

**Figure 7.** Vertical profiles at 1800UTC in the point indicated in figure 6b for the run with the flux-conservative interface. (a) precipitation fluxes: rain (black solid line), snow (red dashed line) and graupel (green dash-dotted line); (b) cold pool-generating phenomena: latent heat effects due to phase changes (black solid line) and sensible heat advection (red dashed line); (c) same as (b) but focused on near-surface.

dynamics interface that makes some ad-hoc approximations. By moving to the generalized flux-conservative equations, the effect of these approximations can be studied.

Monthly verification scores show that the overall effect of introducing the flux-conservative equations in AROME is quite limited. There is no significant improvement or degradation of these scores. Given the mentioned theoretical benefits of the
presented equations, this means that the presented work is a valuable advancement of the AROME model. Moreover, it appears that substantial differences may exist in specific cases. A detailed study of a heavy-precipitation case gives the example of the formation of a cold-pool, which is an essential mechanism in the life-cycle of a convective event. As it appears, one mechanism that contributes to the formation of this cold pool is the heat transport by precipitation. This effect is neglected in the existing AROME physics-dynamics interface, while it is correctly accounted for in the presented flux-conservative set of equations. In
this specific case, this leads to a different surface temperature and surface pressure within the cold pool. A more systematic study of the effect of heat transport on the life-cycle of a cold-pool is left for future research. In this manuscript, this case serves as an illustration of the importance of correctly accounting for supposedly small terms in the energy budget, something that is achieved with the presented set of thermodynamic equations.

Besides offering a direct improvement of the thermodynamic budget of the physics parameterizations of the AROME model,
the presented set of equations also paves the way for interesting future research. Especially the impact of the heat from physics parameterizations on the continuity equation, and the effect of accounting for the net mass exchange between the atmosphere and the surface, are topics that deserve to be studied in detail.

**Code availability**

The used ALADIN Codes, along with all related intellectual property rights, are owned by the Members of the ALADIN consortium. Access to the ALADIN System, or elements thereof, can be granted upon request and for research purposes only.

*Acknowledgements.* The authors of this paper wish to commemorate Jean-François Geleyn, who in his unique vision and understanding ceaselessly stressed the importance of this topic.

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
