# Peer review of "Generalization and application of the flux-conservative thermodynamic equations in the AROME model of the ALADIN system"

_Geoscientific Model Development, 2015_

## Referee Comment (RC1) · Anonymous Referee #1 · 3 Feb 2016

This is an interesting and useful study on interactions of the physical parametrizations and and model dynamics in the mesoscale NWP model ALADIN-AROME. The manuscript reports about generalisation of the formulations suggested by Catry et al., 2007 and testing them in AROME framework. The time evolution of the specific mass of hydrometeor species and heat content ($c_p T$) due to the physical parametrizations is expressed in a universal compact form as the vertical divergence of the sum of terms related to the fluxes of precipitation, phase conversions of water (pseudofluxes), diffusion and radiation (Eqs. 12 and 13). It is shown that in terms of the standard validation the results show a neutral impact, but may be significant in specific (convection) cases. In any case, the suggested formulation allows consistent and systematic building of

the interfaces between model physics and dynamics, by default conserving energy and mass during the model integration.

A general comment is that the tone of the manuscript is somewhat apologising, defendig. The authors feel that the influence of the suggested formulations on weather forecast is small and try to convince the reader that in spite of this it should still be applied in the operational NWP model. They do not discuss systematically possible problems related to the introduction of the new formalism, but a feeling remains that such may exist as they try kind of convince invisible opponents? Based on the information given in the manuscript it is evident that the system of equations if physically logical and well based, allows using less simplifying assumptions and a clear definition of the remaining ones, conserves energy and mass. If, in addition, the application of the new approach does not lead to worse weather forecast (neutral impact), computational cost does not increase significantly, the model code does not grow extremely complicated etc, then it seems natural that it should be applied. In this context, it might be good to mention the limitations of the standard station verification applied to validate research results against surface pressure, screen-level temperature and relative humidity and anemometer-level wind when the verification is

In this sense, the key component seems to be formulation of the pseudofluxes, related to the phase changes between the chosen prognostic hydrometeor species, water vapour (and dry air). It would be good to discuss concepts related to these fluxes, perhaps using an example of some of the existing and/or not yet introduced $R\_j$ terms (conversion between cloud ice crystals and precipitating snow, cloud liquid droplets and cloud ice crystals, for example). There is most probably a whole system of microphysics parametrizations behind any of these terms. Not all details should be discussed but the principles discussed and references given, to allow understanding the concepts not only at the level of general formalism.

Another issue are the limitations of the approach. Three limitations are mentioned in the text but not discussed systematically: 1) interactions between microphysics related to the radiation transfer, on one hand, and that related to the evolution of clouds and precipitation, on the other hand, 2) interactions related to the surface, e.g. between moisture in air and in the soil, 3) the presented formulations being based on the hydrostatic assumption. In this case, the equations were evidently applied in the non-hydrostatic AROME. However, no details are given, but mentioned that it is safe to apply the hydrostatic equations also in the nonhydrostatic model (p.7,l.19). In the conclusions, the possibility to approach in the future these now missing interactions related to radiation and surface could be outlined.

The manuscript is generally well structured and written. It makes understandable features, possibly remaining unclear by reading the original Catry et al, 2007 paper. However, a lot of details should be presented more carefully as now concepts and variables are discussed before they are defined and some essential information seems to be missing. The English language is understandable to me but I would leave the Editor to judge if there is room for improvement.

My specific comments are given as comments in the manuscript pdf (made with Acrobat Reader). Hopefully, their contents are visible by using the same reader.

The manuscript could be approved after taking into account these general and specific comments. In my opinion, this requires at least a medium-size revision.

Please also note the supplement to this comment:
http://www.geosci-model-dev-discuss.net/gmd-2015-279/gmd-2015-279-RC1-supplement.pdf

**Supplement:**

[revised manuscript text omitted]

---

## Referee Comment (RC2) · Anonymous Referee #2 · 4 Feb 2016

The study constructs a general form of the flux-conservative physics-dynamics interface described in Catry et al., 2007, and replaces the temperature-tendency based interface in AROME with this new interface.

The new interface offers an energy conserving system, and provides a more accurate model. It provides an opportunity to import new physical parameterizations into the energy conserving frame-work, and illustrates an example where the advection of sensible heat of precipitation can be of importance for the forecast.

The study highlights a topic (physics-dynamics interface) which is often overlooked. In particular in the climate community, much more attention ought to be paid on an energy conserving system, and this is a very nice contribution demonstrating how it can be

applied in a general sense. It is written in a clear way, and I recommend publication upon a few minor edits.

Page 3: Line 14: relaxation time Bott (2008) → relaxation time, Bott (2008).

Page 3: Line 17: heats → heat

Page 3: Line 18: heats → heat

Page 4: Line 24: heats → heat, appear in → appear on

Page 4: Line 25: these latent heats → the latent heat

Page 5: Line 5:(Piriou et al., 2007), . . . Also → (Piriou et al., 2007). Also

Page 5: Line 7: rain condensation → rain evaporation (?)

Page 5: Line 8: all kinds of transfers → all kind of transfers

Page 5: Line 11: It would be helpful to define j on this line before it is used, e.g "k=0 denotes the dry air component, and j denotes the the conversion process."

Page 5: Line 11: heat capacities → heat capacity

Page 5: Line 15: if you define j further up, remove "in the conversion process j" from this line.

Page 8: Section 3. It would be clearer if you changed the word "current" to "previous" or something alike everywhere in this section. The first time I read through the section I thought it explained the new interface in the "current study", which was confusing.

Page 10: Line 2: Meteorologic → Meteorological

I have opened the .pdf in two different versions of acrobate reader, but the equations are not lining up, and in some places there are symbols such as diamonds and circles, I am not sure if this is a problem on my end with an old version of adobe, or if it is something that should be addressed.

---

## Referee Comment (RC3) · Anonymous Referee #3 · 8 Feb 2016

This manuscript focuses on an energetically and mass consistent physics-dynamics interface and is an extension of a previous paper by Catry et al (2007) in that it allows for an arbtitrary number of air constituents and their interactions.

This type of work is very welcome for the scientific community as it aims at standardising general physics-dynamics interfaces.

The manuscript is divided in a more theoretical part an a more applied part.

Regarding the theoretical part, as far as I could see, the statements given are all true and useful. The equation set (2-8) is only the equation set for the physics, not for the whole model. This should be made clear. It is not clear why pseudo-fluxes are

employed to describe source terms. I think that this makes the issue unecessarily unintuitive. Why the flux-conservative form is enforced here? Is there a coding style advantage?

Regarding the sedimentation fluxes in equations (9) and (10), I can't see at a glance why the rain flux Pr shoud depend on both absolute Pr* and Ps*. Could you explain this?

You also mention the relative flux of dry air to be defined as Pd=-sum(Pk) (Page 7 about lines 10). This is correct. What is about the flux of water vapour or other non-sedimenting species? It should have the same compensating velocity as dry air.

In Section 4 it is mentioned that the surface boundary condition of AROME does not allow for mass exchange between soil and atmosphere. A consequence is then, that energy exchange associated with moisture and precip is also not possible? Do I understand this correctly? Then, with regard to the cold pool example you give later on in Figure 7, which consequences would this imply? Could you try to implement this boundary condition? An why should it not be possible to implement this boundary condition?

Even if one might believe that the endeavour to introduce more consistency does not result in better overall scores, the cold pool example shows clearly the advantages in extreme weather situations. This is more than sufficient.

---

## Referee Comment (RC4) · Anonymous Referee #4 · 10 Mar 2016

**"Generalization and application of the flux-conservative thermodynamic equations in the Arome model of the Aladin system",**
**D. Degrauwe, Y. Seity, F. Bouyssel, P. Termonia**

**1   General comments**

Ensuring the consistency of the coupling between a dynamical core and a physics package is a delicate matter when building a NWP model from two blocks which have been developed independently. This article present a new physics/dynamics interface which has been designed to interface the Aladin non-hydrostatic dynamical core with the convection permitting physics package originally developed for the research model Meso-NH. The new interface which is a generalization of a concept proposed by Catry et al, 2007, is supposed to improve mass and energy conservation and thermodynamical consistency between the two packages.

The authors claim that, thanks to the equations used in this interface "the conservation of mass and energy is a built-in feature of the system". I don't agree with this affirmation. I think the problem is much more complicated, and the best solution for the coupling depends a lot of the method which has been chosen inside the physics driver and inside the parametrisations themselves. I don't think this interface replaces a global solver which would be the only clean way to solve consistently together all the processes represented by the system of equations (2)-(8).

I think that what is really missing in this paper is a proper discussion about the discretization of the complex system given by equations (2)-(8). The only example which is given in page 8, line 12 is actually not valid for any physics package. In section 3 below, I illustrate with a simple example extracted from the paper why I don't think that the conservation of mass and energy is necessarily a built-in feature of the system of equations (2)-(8).

I don't think either that for local processes such as autoconversion or condensation, pseudo-fluxes are necessary to ensure conservation. The pseudo-fluxes are not necessary to express the system in a barycentric form. If a parametrisation has not been written in term of pseudo-fluxes but the parametrisation gives tendencies for the $q_x$, how should the pseudo-fluxes be computed? Should for example the method take into account the fact that the latent heat used in a parametrisation is $L(T)$ instead of $L(T_o)$?

The test case of section 4 shows that the main impact of the new interface has been to include a term which was missing (had been neglected?) in the parametrisation of the precipitation sedimentation. From the text, it is not clear if adding this term in the parametrisation, but still using the old interface, would produce the same effect. Could the author try to separate more clearly the impact of the new design of the interface from the impact of the missing term?

I also think that the title of the paper is very misleading. The formulation of the Aladin dynamical core is not based on conservative flux-form equations. The new interface

will surely not improve the non-conservative aspects of the Aladin semi-Lagrangian advection scheme for example. In the paper, conservative flux-form equations are used only to compute the tendencies from information provided by the physics parametrisations. This should be made clearer in the title and in the abstract.

I don't think there is a lot new information in the current state of the manuscript about physics/dynamics interface compared to Catry et al (2007) (the generalization to more water species and processes is quite straightforward). A more careful and general analysis of the physics/dynamics interfacing problem should be added to the manuscript to make it useful to the community. The results concerning the impact of the missing term in the sedimentation of precipitation are interesting but only more systematic sensitivity tests and comparison to observation would prove the importance of this effect versus many other sources of model errors for the simulation of organized convection in convection permitting NWP.

**2  More detailed comments**

1. p 2, l3 : It is not correct to say that the dynamical core equations are equations written for a perfect gas. The equations of the Aladin dynamical core are written for a "barycentric" multiphase system which may contain condensed water phases, even if the physics is switched off. The water vapor but also the condensed species are taken into account to compute the mass of air in a given volume (e.g. liquid and solid species are "loading" the air parcel) and are then changing the "inertia" of the air parcel in the momentum equation. The gas law also knows about the composition of the air parcel, and only the "gas" part of the total mass is used in the gas law. But the full weight is used for the hydrostatic equilibrium. The information about the composition of the air parcels is known thanks to the definition of the specific water contents which are defined as the ratio between the mass of a given species and the total mass (including condensates). The specific quantities are used to compute the virtual temperature and the moist $c_p$ (thermal inertia) and the moist gas constant $R$ which are also used inside the dynamics.

2. p 2, l1-2 : Why only mass and enthalpy budgets but no momentum budget in this interface ?

3. p2 l25 : I don't really understand this sentence. Shouldn't it be "to be described" instead of "to described" ?

4. in section 2.1, the authors list the main hypotheses made for the design of Catry et al (2007) interface. However, nothing is said about the "Eulerian" and "vertical column" hypotheses used to write the system of equation (2)-(8) and how such a system should be interfaced with the dynamics. For example, in the case of a semi-Lagrangian dynamics, should the tendencies be computed at the beginning/middle/end of the semi-Lagrangian trajectories or along the trajectories ?

5. p3 l7-10 : It is not clear if the precipitation are supposed to immediately get the

temperature of the layers they are crossing during a time step, or only the temperature of the layer where they "seat" at the end of the time step (in other word, is there an exchange of energy with all layers which are crossed by the condensed phases or only inside the layer where they stop at the end of the time step).

6. p3 l8 : Bott (2008) → (Bott, 2008)

7. p3 l 21 : it should be said more clearly that the $q_x$ are specific fractions (i.e. ratios with respect to total mass of the multiphase system).

8. p4 l8 : I understand that in a barycentric system, the total mass should be conserved, i.e the sum of eq 2-7 gives $0 = 0$. Does it mean that the surface scheme should produce diffusive fluxes of condensates to compensate the diffusive flux of water vapor ?

9. p4 l20 : center of mass of what ?

10. p7 l12-l15 : "all vertical transport is compensated by a flux of dry air" : I don't think it applies to "all vertical transport" (in particular, it does not applied to resolved vertical transport), but only to precipitation and subgrid mass transports (top of page 5 of Courtier et al, 1991).

11. p8 l7-12 : Formulae p8, l12 ensure conservation only if the parametrisations are called in parallel (process split in Williamson, 2002). It also supposes that the final specific ratios are given by process (a) and (b) (and not by a common resolution of both process (a) and (b)). If the parametrisations are called sequentially (time split), parametrisation (b) will already know about the evolution in time of both $T$ and $c_p$ after process (a). In this case, the conservation is ensure if $\Delta T = \Delta T_a + \Delta T_b$. See annexe for details.

12. p9, l31-32 and p10 l1-2 : The authors say that, in Arome, it is not possible to take into account the correct mass budget, therefore all vertical transport is compensated by a flux of dry air. Does it mean that equation 43 of Catry et al is used instead of equation (8) in the new interface ? Why the correct mass budget could not be taken into account in Arome ? Is the problem only at the surface or at every level ?

13. p10 l17 : I don't clearly understand why no significant improvement could be expected without a new tuning. Are the author thinking of model error compensation ? If it is the case, it should be explained more clearly.

14. section 4.2 : the discussion for this case study mainly consider the missing term (heat transport by the precipitation). If this term is the main problem in Arome, couldn't it be added to the old interface ? Would it give similar results ?

15. It would also be interesting to see the impact of the new interface independently from the addition of the missing term for the case in section 4.2 (i.e. also neglect the missing term in the new interface).

16. The author should be more modest in their conclusion. The simulation of such convective systems are very sensitive to many other source of model errors (time step, horizontal or vertical resolution, level of complexity in microphysics etc). A

more systematic study would be necessary to really conclude about the importance of the heat transport by precipitation in Arome.

**3    Annexe**

One of the "moist" quantity which must be conserved in parametrisations involving "warm" water phase changes (pseudo-fluxes) is $s = c_p T + L q_v$ where $c_p = (1 - q_v - q_l) c_{p_d} + q_v c_{p_v} + q_l c_l$ and $L = L(T_{00})$ is the latent heat of vaporisation at $T_{00} = 0$ K.

At time $t$, $s^0 = c_p^0 T^0 + L q_v^0$.

If the process (a) is conservative :

$$s^0 = c_p^a T^a + L q_v^a = (c_p^0 + \Delta c_p^a)(T^0 + \Delta T^a) + L(q_v^0 + \Delta q_v^a)$$

Process (b) is "called" in parallel and (b) is also a conservative process :

$$s^0 = c_p^b T^b + L q_v^b = (c_p^0 + \Delta c_p^b)(T^0 + \Delta T^b) + L(q_v^0 + \Delta q_v^b)$$

At the end of the physics (i.e. after (a+b) here), we have conservation if

$$s^0 = c_p^+ T^+ + L q_v^+$$

If we keep the water phase changes as given by processes (a) and (b), i.e.

$$c_p^+ = (1 - q_v^+ - q_l^+) c_{p_d} + q_v^+ c_{p_v} + q_l^+ c_l$$

with $q_v^+ = q_v^0 + \Delta q_v^a + \Delta q_v^b$ and $q_l^+ = q_l^0 - \Delta q_v^a - \Delta q_v^b$ we get

$$T^+ = \frac{1}{c_p^+} \left( c_p^0 T^0 + L q_v^0 \right) - L q_v^+$$

after a bit of arithmetics, we get (using the conservation for each process) :

$$T^+ - T_0 = \Delta T^+ = \frac{(c_p^0 + \Delta c_p^a)(\Delta T^a) + (c_p^0 + \Delta c_p^b)(\Delta T^b)}{c_p^+}$$

i.e. formulae p8, line 12.

Note that, in this case, if $T_{a/b}$ and $q_{v_{a/b}}$ have been computed in the parametrisations (a/b) in order to fulfill some kind of fast adjustment towards an equilibrium, the adjustment will not be valid anymore with $T^+$.

Now, if process (b) is "called" sequentially after (a) (and (b) is still a conservative process) :

$$s^0 = c_p^b T^b + L q_v^b = (c_p^0 + \Delta c_p^a + \Delta c_p^b)(T^0 + \Delta T^a + \Delta T^b) + L(q_v^0 + \Delta q_v^a + \Delta q_v^b)$$

In this case, we directly get :

$$T^+ - T^0 = \Delta T^a + \Delta T^b$$

In this case, if $T_b$ and $q_{v_b}$ have been computed in order to fulfill some kind of fast adjustment towards an equilibrium, the adjustment will still be valid with $T^+$.

---

## Author Comment (AC1) · 13 Apr 2016

**Author's response to Referee #1**

**Referee's comment:** *A general comment is that the tone of the manuscript is somewhat apologising, defendig. The authors feel that the influence of the suggested formulations on weather forecast is small and try to convince the reader that in spite of this it should still be applied in the operational NWP model. They do not discuss systematically possible problems related to the introduction of the new formalism, but a feeling remains that such may exist as they try kind of convince invisible opponents?*

The authors admit that they have been very careful in their formulations. This general comment aside, we agree that the virtues of the approach presented in this manuscript could be accentuated more. (see also next comment)

**Referee's comment:** *Based on the information given in the manuscript it is evident that the system of equations if physically logical and well based, allows using less simplifying assumptions and a clear definition of the remaining ones, conserves energy and mass. If, in addition, the application of the new approach does not lead to worse weather forecast (neutral impact), computational cost does not increase significantly, the model code does not grow extremely complicated etc, then it seems natural that it should be applied.*

We obviously believe in the value of this work, and we are grateful that it is acknowledged by this referee. We share the referee's view that the advantages of the presented set of equations are sufficient to justify their application (given the neutral impact etc.). A sentence to emphasize this will be added to the conclusions of the manuscript.

**Referee's comment:** *In this context, it might be good to mention the limitations of the standard station verification applied to validate research results against surface pressure, screen-level temperature and relative humidity and anemometer-level wind when the verification is* (sentence truncated in Referee's document)

This is a good suggestion. Implicitly, the manuscript already points out the limitations of standard verification by the inclusion of a case-study, but we agree that it should be stated explicitly. A paragraph has been added to the manuscript to amend this.

**Referee's comment:** *In this sense, the key component seems to be formulation of the pseudofluxes, related to the phase changes between the chosen prognostic hydrometeor species, water vapour (and dry air). It would be good to discuss concepts related to these fluxes, perhaps using an example of some of the existing and/or not yet introduced R_j terms (conversion between cloud ice crystals and precipitating snow, cloud liquid droplets and cloud ice crystals, for example). There is most probably a whole system of microphysics parametrizations behind any of these terms. Not all details should be discussed but the principles discussed and references given, to allow understanding the concepts not only at the level of general formalism.*

The concept of pseudofluxes is indeed the key to the flux-conservative formulation. However, this does not mean that the physical parameterizations should use these pseudofluxes internally. In fact, the AROME microphysics are internally formulated in terms of local tendencies $\partial q_k / \partial t$. In order to use the flux-conservative equations, these tendencies are then converted to pseudofluxes with the formula

$$R_j = \int_0^p \frac{1}{g} \frac{\partial q_k}{\partial t} dp \tag{1}$$

The referee is entirely right that the manuscript is lacking a more detailed explanation of why

pseudofluxes are introduced, and how they relate to (the more conventional) local tendencies. A section will be added to the manuscript to remediate this.

As the referee indicates, the internal machinery of the (microphysical) parameterizations that provide the pseudofluxes is quite complex and a topic in itself. References to such parameterizations are already given in the manuscript, e.g. Lascaux et al. (2007). The authors therefore believe that the discussion of the internals of such schemes falls outside the scope of the presented manuscript.

**Referee's comment:** *Another issue are the limitations of the approach. Three limitations are mentioned in the text but not discussed systematically: 1) interactions between microphysics related to the radiation transfer, on one hand, and that related to the evolution of clouds and precipitation, on the other hand, 2) interactions related to the surface, e.g. between moisture in air and in the soil, 3) the presented formulations being based on the hydrostatic assumption. In this case, the equations were evidently applied in the non-hydrostatic AROME. However, no details are given, but mentioned that it is safe to apply the hydrostatic equations also in the nonhydrostatic model (p.7,l.19). In the conclusions, the possibility to approach in the future these now missing interactions related to radiation and surface could be outlined.*

1. Interactions between the microphysics parameterization and the radiation parameterization are not a limitation of the set of equations itself. The scope of the presented manuscript is the interaction between the parameterizations and the dynamical core, but *not* the interaction between one parameterization and another. To indicate that such interactions exist, the example is given of cloud microphysics influencing radiation. The scope limitations of the present work will be mentioned more explicitly in the introduction section.

2. The referee points out correctly that the interaction with the surface is somewhat underexposed in the manuscript. Regarding this topic, we would like to make the following remarks:

   - The presented set of equations describes the impact of physical parameterizations on atmospheric prognostic variables. In this sense, the evolution of the (sub)surface prognostic variables falls outside the scope of this document.
   - The presented flux-based equations do not pose any direct limitations regarding the interaction with the surface. Moreover, they are directly compatible with the work of Best et al. (2004), who present a generalized flux-based coupling between the surface scheme and the atmospheric model. The AROME model uses the interface from Best et al. (2004) to couple its externalized surface scheme SURFEX to the upper-air parameterizations.

   These points are added to the manuscript.

3. The issue of applying the presented set of equations in a non-hydrostatic model like AROME is discussed in section 2.4. The following 2 points are made there:

   - The extension of the flux-based equations to the non-hydrostatic case is pretty straightforward, as is shown in Catry et al. (2007). The main consequence of this extension is that heat appears as a source not only in the temperature equation, but also in the continuity equation.
   - Evidence is found in literature that the impact of the source term in the pressure equation is quite limited. This is not just 'mentioned' in the manuscript, but a reference is given.

   Given these points, we have chosen to use the 'hydrostatic formulation' in the nonhydrostatic AROME model. Maybe this was not sufficiently clear from the manuscript, so it has been rephrased somewhat in section 2.4, and it is repeated in section 4. A sentence is also added in the conclusions to mark an in-depth assessment of using the nonhydrostatic formulation of the flux-based thermodynamic equations as an interesting future research topic.

**Referee's comment:** *My specific comments are given as comments in the manuscript pdf*

We agree with most of these valuable suggestions. The manuscript will be adapted accordingly.

---

## Author Comment (AC2) · 13 Apr 2016

**Author's response to Referee #2**

**Referee's comment:** *It is written in a clear way, and I recommend publication upon a few minor edits.*

*Page 3: Line 14: relaxation time Bott (2008) → relaxation time, Bott (2008).*

*Page 3: Line 17: heats → heat*

*Page 3: Line 18: heats → heat*

*Page 4: Line 24: heats → heat, appear in → appear on*

*Page 4: Line 25: these latent heats → the latent heat*

*Page 5: Line 5:(Piriou et al., 2007), . . . Also → (Piriou et al., 2007). Also*

*Page 5: Line 7: rain condensation → rain evaporation (?)*

*Page 5: Line 8: all kinds of transfers → all kind of transfers*

*Page 5: Line 11: It would be helpful to define j on this line before it is used, e.g k=0 denotes the dry air component, and j denotes the the conversion process.*

*Page 5: Line 11: heat capacities → heat capacity*

*Page 5: Line 15: if you define j further up, remove in the conversion process j from this line.*

*Page 8: Section 3. It would be clearer if you changed the word current to previous or something alike everywhere in this section. The first time I read through the section I thought it explained the new interface in the current study, which was confusing.*

*Page 10: Line 2: Meteorologic → Meteorological*

The authors agree with most of these comments. The manuscript is adopted accordingly.

---

## Author Comment (AC3) · 13 Apr 2016

**Author's response to Referee #3**

**Referee's comment:** *The equation set (2-8) is only the equation set for the physics, not for the whole model. This should be made clear.*

The scope of the presented manuscript is indeed restricted to the equation set of the physics, something that was not sufficiently clear in the original manuscript.

The introduction is modified in order to set this scope more clearly. Also at the point where the equations are introduced, it is now stated explicitly that these equations only describe the physics part. The abstract and conclusions section also have been modified to emphasize this important restriction of the scope.

**Referee's comment:** *It is not clear why pseudo-fluxes are employed to describe source terms. I think that this makes the issue unecessarily unintuitive. Why the flux-conservative form is enforced here? Is there a coding style advantage?*

Writing equations in a flux-conservative formulation is a way to ensure that conservation laws are obeyed after spatial and temporal discretization. This technique is applied commonly in the dynamics (literature references are given in the introduction section), and in this manuscript it is applied in the physics.

In order to write the physics equations, and more specifically the contributions of phase changes, in a flux-conservative way, the concept of pseudofluxes is introduced in Catry *et al.,* (2007) . We agree that this concept deserves more explanation, and the manuscript is adopted in this sense. Also, it is noted that this does not mean that the internals of the physics parameterizations should be formulated in terms of pseudofluxes. Instead, it is quite easy to calculate pseudofluxes from local species tendencies by integrating from the top of the atmosphere:

$$R_j = \int_0^p \frac{1}{g} \frac{\partial q_k}{\partial t} dp \tag{1}$$

(same notations as in the manuscript).

This equation is also added to the manuscript.

**Referee's comment:** *Regarding the sedimentation fluxes in equations (9) and (10), I cant see at a glance why the rain flux Pr shoud depend on both absolute Pr\* and Ps\*. Could you explain this?*

The relative precipitation flux $P_r$ describes the movement of rain with respect to the mass center of the parcel, including *all* components. It may be easier to understand when writing the fluxes explicitly as a product of (vertical) velocity and specific humidity:

$$P_r^* = \rho_r w_r^* = \rho_{tot} q_r w_r^* \tag{2}$$
$$P_s^* = \rho_s w_s^* = \rho_{tot} q_s w_s^* \tag{3}$$

where $\rho_r$, $\rho_s$ and $\rho_{tot}$ are the rain density, snow density and total density, and $w_r^*$ and $w_s^*$ are the absolute velocities of rain and snow. If we now consider a parcel only consisting of dry air, rain and snow, the absolute velocity $w^*$ of the center of mass of this parcel is equal to

$$w^* = w_r^* q_r + w_s^* q_s \tag{4}$$

The relative velocities of rain and snow then become

$$w_r = w_r^* - w^* \tag{5}$$
$$w_s = w_s^* - w^* \tag{6}$$

and the relative precipitation fluxes become

$$P_r = \rho_{tot} q_r w_r = \rho_{tot} q_r (w_r^* - w^*) = (1 - q_r)P_r^* - q_r P_s^* \qquad (7)$$

$$P_s = \rho_{tot} q_s w_s = \rho_{tot} q_r (w_s^* - w^*) = (1 - q_s)P_s^* - q_s P_r^* \qquad (8)$$

as given in the manuscript.

Some explanation is added to the manuscript to clarify the relationship between absolute and relative precipitation fluxes.

**Referee's comment:** *You also mention the relative flux of dry air to be defined as Pd=-sum(Pk) (Page 7 about lines 10). This is correct. What is about the flux of water vapour or other nonsedimenting species? It should have the same compensating velocity as dry air.*

The mentioned equation ($P_d = -\sum_k P_k$) only holds when the approximation is made that all mass transport by physical processes is compensated for by a fictitious flux of dry air. The fact that this is a rather artificial approximation is indeed insufficiently clear from the manuscript. The text has been modified to amend this.

Regarding water vapour and nonsedimenting species: under this approximation, they do not have the same vertical velocity as dry air; in fact, their vertical velocity is zero. In other words, the behavior of dry air is rather artificial (relatively large vertical velocity), but the description of the transport of the suspended water species is close to what happens in the atmosphere. It should be noted that this artificial behavior of dry air is restricted to the physics-dynamics interface: it does not affect the dynamical equations, nor does it affect the internals of the physics parameterizations.

**Referee's comment:** *In Section 4 it is mentioned that the surface boundary condition of AROME does not allow for mass exchange between soil and atmosphere. A consequence is then, that energy exchange associated with moisture and precip is also not possible? Do I understand this correctly? Then, with regard to the cold pool example you give later on in Figure 7, which consequences would this imply? Could you try to implement this boundary condition? An why should it not be possible to implement this boundary condition?*

This is a correct remark by the referee. The statement that mass exchange is not allowed is not entirely accurate. It should rather be that there is no *net* mass exchange. In other words, all mass exchange of water at the surface (evapotranspiration and precipitation) is compensated for by a fictitious exchange of dry air.

Therefore, water mass exchange is possible –and accounted for– in AROME, as are the thermodynamic effects of these processes.

The reason that it is quite difficult to implement this boundary condition in AROME is that the vertical coordinate of the AROME model is mass-based. Correctly accounting for a net mass exchange between atmosphere and surface would have far-reaching consequences, especially in the solution of the dynamical equations. We agree with the referee that this was not explained sufficiently in the manuscript, and some sentences are added to amend this. This issue is also mentioned in the conclusions section as an interesting future research topic.

---

## Author Comment (AC4) · 13 Apr 2016

**Author's response to Referee #4**

**1 General comments**

**Referee's comment:** *The authors claim that, thanks to the equations used in this interface "the conservation of mass and energy is a built-in feature of the system". I don't agree with this affirmation. I think the problem is much more complicated, and the best solution for the coupling depends a lot of the method which has been chosen inside the physics driver and inside the parametrisations themselves. I don't think this interface replaces a global solver which would be the only clean way to solve consistently together all the processes represented by the system of equations (2)-(8).*

Writing equations in a flux-based formulation is in fact a fairly common way to enforce conservation laws. Several references are given in the manuscript to show this. Could the referee please explain why this technique would not lead to a conserving system when applied in the field of physics-dynamics coupling?

**Referee's comment:** *I think that what is really missing in this paper is a proper discussion about the discretization of the complex system given by equations (2)-(8). The only example which is given in page 8, line 12 is actually not valid for any physics package. In section 3 below, I illustrate with a simple example extracted from the paper why I don't think that the conservation of mass and energy is necessarily a built-in feature of the system of equations (2)-(8).*

The referee is entirely correct to point out that the resulting equation of the example given in page 8 is not valid for a time-split (sequential) coupling case. The fact that this example concerns the case of a process-split (parallel) coupling should definitely be stated explicitly in the manuscript. In the case of a time-split coupling, the equations (2)-(8) [or (12)-(13)] are still valid, but the discretization looks somewhat different.

Let's resume the same example of two parameterizations (denoted $(a)$ and $(b)$), but this time with a time-split coupling. Assuming that each process is energy-conserving in itself means that the change in enthalpy due the first process is given by

$$\Delta h^a = (c_p^t + \Delta c_p^a)(T^t + \Delta T^a) - c_p^t T^t \tag{1}$$

To determine the enthalpy change due the second process, we should account for the fact that in a time-split coupling, it doesn't start from $c_p^t$ and $T^t$, but from $\tilde{c}_p = c_p^t + \Delta c_p^a$ and $\tilde{T} = T^t + \Delta T^a$. Therefore, the change in enthalpy due to the second process is given by

$$\Delta h^b = (\tilde{c}_p + \Delta c_p^b)(\tilde{T} + \Delta T^b) - \tilde{c}_p \tilde{T} \tag{2}$$

The combined effect of both parameterizations is then written as

$$\Delta h = \Delta h^a + \Delta h^b = c_p^{t+\Delta t} T^{t+\Delta t} - c_p^t T^t \tag{3}$$

from which the temperature at the next timestep is solved as

$$T^{t+\Delta t} = \frac{c_p^t T^t + \Delta h}{c_p^{t+\Delta t}} = T^t + \Delta T^a + \Delta T^b \tag{4}$$

i.e. the same result as given by the referee in the Annexe. This shows that, when applied correctly, our set of equations is also valid for time-split coupling.

We agree with the referee that some explanation about the application of the presented equations in the case of a time-split coupling is missing in the manuscript, and a section has been added on

this issue. However, we do not agree on the point that the conserving property of the equations does not hold in the case of a time-split coupling.

The remark of the referee about the atmospheric state not being adjusted after a process-split coupling is correct. However, this is a problem inherent to process-split coupling, not to the presented set of equations.

**Referee's comment:** *I don't think either that for local processes such as autoconversion or condensation, pseudofluxes are necessary to ensure conservation. The pseudofluxes are not necessary to express the system in a barycentric form. If a parametrisation has not been written in term of pseudofluxes but the parametrisation gives tendencies for the qx, how should the pseudofluxes be computed ? Should for example the method take into account the fact that the latent heat used in a parametrisation is $L(T)$ instead of $L(T_0)$ ?*

The concept of pseudofluxes is indeed insufficiently explained in the manuscript. The referee is correct when saying that it is not necessary to ensure conservation: if a physics parameterization is constructed properly, it should conserve mass and energy in itself. However, the pseudofluxes are a way to enforce conservation at a higher level, for example to make sure that energy is also conserved when several parameterizations are put together (see previous comment). Writing the combined effect of all parameterizations in a single flux-conservative equation is only possible through the concept of pseudofluxes.

The microphysical parameterizations of AROME are internally formulated in terms of specific humidity tendencies. The transformation of these tendencies to pseudofluxes is simply done by taking a vertical integral:

$$R_j = \int_0^p \frac{1}{g} \frac{\partial q_k}{\partial t} dp \tag{5}$$

This equation and method are added to the manuscript.

The transformation of tendencies into fluxes does not depend on whether $L(T)$ or $L(T_0)$ is used. It is important to keep in mind that the pseudofluxes only describe a mass exchange between different species. The parameterizations should not determine the thermodynamic effect of phase changes themselves (well, they can do it for internal purposes, but they should not pass this information to the dynamical core). Instead, they only determine the effect of the phase changes on the specific humidities (expressed through pseudofluxes), and equation (13) of the manuscript determines its thermodynamic effect.

**Referee's comment:** *The test case of section 4 shows that the main impact of the new interface has been to include a term which was missing (had been neglected ?) in the parametrisation of the precipitation sedimentation. From the text, it is not clear if adding this term in the parametrisation, but still using the old interface, would produce the same effect. Could the author try to separate more clearly the impact of the new design of the interface from the impact of the missing term ?*

It has been tested carefully what the origin is of the differences between results with the temperature-based interface and with the flux-based interface. This has led to the identification of the list of approximations that is given in section 3. The referee is correct when saying that the effect of heat transportation by precipitation could also be included in the temperature-based interface.

However, this is not the point we wish to make with this test case. The purpose of this case is merely to show that small terms in the energy budget, which can safely be neglected on a large scale, still can have a significant impact under specific circumstances. The use of the eqs. (12)-(13) avoids such approximations. This is stated clearly in the last paragraph of section 4.

**Referee's comment:** *I also think that the title of the paper is very misleading. The formulation of the Aladin dynamical core is not based on conservative flux-form equations. The new interface will surely not improve the non-conservative aspects of the Aladin semi-Lagrangian advection scheme for example. In the paper, conservative flux-form equations are used only to compute the tendencies from information provided by the physics parametrisations. This should be made clearer in the title and in the abstract.*

The first sentence of the abstract already defines the scope of this work as the "thermodynamic impact of physical parameterizations". However, we agree that it is quite important that the reader keeps this in mind. Therefore, the fact that only the effect physical parameterizations is considered, is added explicitly at several places in the manuscript: (i) in the abstract, (ii) in the introduction section, (iii) at the point where the equations are introduced, and (iv) in the conclusions.

**Referee's comment:** *I don't think there is a lot new information in the current state of the manuscript about physics/dynamics interface compared to Catry et al (2007) (the generalization to more water species and processes is quite straightforward). A more careful and general analysis of the physics/dynamics interfacing problem should be added to the manuscript to make it useful to the community. The results concerning the impact of the missing term in the sedimentation of precipitation are interesting but only more systematic sensitivity tests and comparison to observation would prove the importance of this effect versus many other sources of model errors for the simulation of organized convection in convection permitting NWP.*

We agree that the generalization is straightforward (from a mathematical point of view). However, this generalization is essential for the application in the AROME model, which is used for operations and/or research in 26 European and North-African countries. The application in AROME has led to the identification of some approximations that are present in the existing temperature-based interface of AROME. We believe that this is very valuable information for the users of this model. Moreover, our work provides a solution to get rid of these approximations, which is also of direct interest to these people.

A more complete investigation of the phenomenon discussed in the case study indeed would be interesting. This remark has been added to the manuscript. But as indicated in the manuscript and in reply to another comment, the phenomenon of heat transportation by precipitation is not the topic of the presented manuscript, but serves as an example of a case where the aforementioned approximations are not harmless.

**2  More detailed comments**

**Referee's comment:** *1. p 2, l3 : It is not correct to say that the dynamical core equations are equations written for a perfect gas. The equations of the Aladin dynamical core are written for a "barycentric" multiphase system which may contain condensed water phases, even if the physics is switched off. The water vapor but also the condensed species are taken into account to compute the mass of air in a given volume (e.g. liquid and solid species are "loading" the air parcel) and are then changing the "inertia" of the air parcel in the momentum equation. The gas law also knows about the composition of the air parcel, and only the "gas" part of the total mass is used in the gas law. But the full weight is used for the hydrostatic equilibrium. The information about the composition of the air parcels is known thanks to the definition of the specific water contents which are defined as the ratio between the mass of a given species and the total mass (including condensates). The specific quantities are used to compute the virtual temperature and the moist $c_p$ (thermal inertia) and the moist gas constant $R$ which are also used inside the dynamics.*

The statement that the dynamical core only considers a perfect gas has been removed from the manuscript.

**Referee's comment:** *2. p 2, l1-2 : Why only mass and enthalpy budgets but no momentum budget in this interface ?*

Exactly the same flux-conservative interface indeed could be used for momentum, as well as for other prognostic variables like TKE, passive tracers, etc. A sentence has been added to the manuscript to point this out. An important difference with water species, is that the evolution of water species affects the evolution of temperature in a complicated way, e.g. through latent heat effects or through the specific heat capacity. Therefore, the application of the presented methodology is much more important and interesting for the case of water species.

**Referee's comment:** *3. p2 l25 : I don't really understand this sentence. Shouldn't it be "to be described" instead of "to described"?*

Indeed, this has been adapted.

**Referee's comment:** *4. in section 2.1, the authors list the main hypotheses made for the design of Catry et al (2007) interface. However, nothing is said about the "Eulerian" and "vertical column" hypotheses used to write the system of equation (2)-(8) and how such a system should be interfaced with the dynamics. For example, in the case of a semi-Lagrangian dynamics, should the tendencies be computed at the beginning/middle/end of the semi-Lagrangian trajectories or along the trajectories?*

The hypothesis framework is indeed restricted to the hypotheses that relate to the thermodynamics of the model. This indeed should be stated more clearly in the manuscript.

The other hypotheses mentioned by the referee are more related to the general design of an atmospheric model. The use of the presented equations is rather independent of these hypotheses. For instance, the Green-Ostrogradsky theorem that forms the basis of the choice for a flux-conservative formulation, is also valid in 3 dimensions, so our work is not only relevant for physics parameterizations that are organized in vertical columns.

**Referee's comment:** *5. p3 l7-10 : It is not clear if the precipitation are supposed to immediately get the temperature of the layers they are crossing during a time step, or only the temperature of the layer where they "seat" at the end of the time step (in other word, is there an exchange of energy with all layers which are crossed by the condensed phases or only inside the layer where they stop at the end of the time step).*

Precipitation also takes the temperature of the layers in-between, and the heat exchange between precipitation and these layers is accounted for. This can be seen from the relevant term in the equation (13) of the original manuscript:

$$\frac{\partial}{\partial t}(c_p T) = -g\frac{\partial}{\partial p}\left[\ldots + \sum_k (c_k - c_p)P_k T\right] \tag{6}$$

The fact that the product of the precipitation fluxes ($P_k$) with temperature ($T$) occurs here, means that there's an energy transfer, even when precipitation just 'falls through' a layer: in such case the divergence of the precipitation flux is zero ($\partial P_k/\partial p = 0$), but the divergence of the heat flux is not necessarily zero ($\partial(P_k T)/\partial p \neq 0$).

**Referee's comment:** *6. p3 l8 : Bott (2008) → (Bott, 2008)*

This has been adapted.

**Referee's comment:** *7. p3 l 21 : it should be said more clearly that the $q_x$ are specific fractions (i.e. ratios with respect to total mass of the multiphase system).*

This has been adapted.

**Referee's comment:** *8. p4 l8 : I understand that in a barycentric system, the total mass should be conserved, i.e the sum of eq 2-7 gives 0 = 0. Does it mean that the surface scheme should produce diffusive fluxes of condensates to compensate the diffusive flux of water vapor?*

Barycentrism doesn't mean that the total mass should be conserved. It means that the motions are considered relative to the (moving) center of mass of the parcel. That the right-hand sides of eqs. (2)-(7) cancel out, is due to the definition of specific humidities: $q_k = \rho_k/\rho_{tot}$. From this, it follows that $\sum_k q_k = 1$, so $\sum_k \partial q_k/\partial t = 0$.

As indicated in Catry *et al.* (2007), a change in total mass due to exchange with the surface, is

reflected in a change of the surface pressure.

**Referee's comment:** *9. p4 l20 : center of mass of what?*

The center of mass of the air parcel. This has been adapted in the manuscript, and a more elaborate explanation of the relation between relative and absolute precipitation fluxes has been added.

**Referee's comment:** *10. p7 l12-l15 : "all vertical transport is compensated by a flux of dry air" : I don't think it applies to "all vertical transport" (in particular, it does not applied to resolved vertical transport), but only to precipitation and subgrid mass transports (top of page 5 of Courtier et al, 1991).*

Indeed, this has been adapted.

**Referee's comment:** *11. p8 l7-12 : Formulae p8, l12 ensure conservation only if the parametrisations are called in parallel (process split in Williamson, 2002). It also supposes that the final specific ratios are given by process (a) and (b) (and not by a common resolution of both process (a) and (b)). If the parametrisations are called sequentially (time split), parametrisation (b) will already know about the evolution in time of both $T$ and $c_p$ after process (a). In this case, the conservation is ensure if $\Delta T = \Delta T^a + \Delta T^b$. See annexe for details.*

A correct remark. The fact that the mentioned formula only holds for process-split coupling has been added to the manuscript. The example also has been extended for the time-split case. (cfr. General comment)

**Referee's comment:** *12. p9, l31-32 and p10 l1-2 : The authors say that, in Arome, it is not possible to take into account the correct mass budget, therefore all vertical transport is compensated by a flux of dry air. Does it mean that equation 43 of Catry et al is used instead of equation (8) in the new interface ? Why the correct mass budget could not be taken into account in Arome ? Is the problem only at the surface or at every level ?*

In the system with 4 hydrometeors of Catry *et al.* (2007), it would indeed be their equation (43) that is used. In the generalized system presented in our manuscript, equation (13) is still valid if a modified relative mass flux of dry air $P_d = -\sum_{k=1}^n P_k$ is used.

The reason that this approximation is necessary is that the vertical coordinate of AROME is mass-based. Hence, correctly accounting for a net mass exchange between atmosphere and surface would have far-reaching consequences, especially for the surface boundary condition of the dynamical core. We agree with the referee that this was not explained sufficiently in the manuscript, and some sentences are added to amend this.

**Referee's comment:** *13. p10 l17 : I don't clearly understand why no significant improvement could be expected without a new tuning. Are the author thinking of model error compensation? If it is the case, it should be explained more clearly.*

Indeed, compensating errors are what we have in mind. This has been added to the manuscript.

**Referee's comment:** *14. section 4.2 : the discussion for this case study mainly consider the missing term (heat transport by the precipitation). If this term is the main problem in Arome, couldn't it be added to the old interface ? Would it give similar results ?*

It could be added to the old interface, and it has been verified that this gives similar results. But this is besides the point of the presented work. The strength of the presented framework lies exactly in the fact that one doesn't have to worry whether one term or another is accounted for in the physics-dynamics interface.

**Referee's comment:** *15. It would also be interesting to see the impact of the new interface independently from the addition of the missing term for the case in section 4.2 (i.e. also neglect the missing term in the new interface).*

As mentioned in response to the previous comment, the value of our work doesn't lie in discussing the impact of individual terms of the energy budget of the AROME model. This is interesting research in itself, but it is not the focus of this manuscript. The focus of this paper rather lies in the presentation of an overall framework that takes care of all terms in a consistent and conserving way.

In this sense, the case of the cold-pool formation under heavy precipitation is just an example where a clear impact of this framework can be noticed. A systematic study of the effect of heat transport of precipitation on the life-cycle of a cold-pool would be interesting in itself, but falls outside the scope of the present work. A sentence has been added to the conclusions to indicate this.

**Referee's comment:** *16. The author should be more modest in their conclusion. The simulation of such convective systems are very sensitive to many other source of model errors (time step, horizontal or vertical resolution, level of complexity in microphysics etc). A more systematic study would be necessary to really conclude about the importance of the heat transport by precipitation in Arome.*

We agree entirely that such a systematic study would be very interesting. But again, it is besides the point we try to make in this manuscript.